# Eruption Pattern of Permanent Canines and Premolars in Polish Children

**DOI:** 10.3390/ijerph19148464

**Published:** 2022-07-11

**Authors:** Katarzyna Cieślińska, Katarzyna Zaborowicz, Zuzanna Buchwald, Barbara Biedziak

**Affiliations:** 1Department of Orthodontics and Craniofacial Anomalies, Poznan University of Medical Sciences, Colegium Maius, Fredry 10, 61-701 Poznan, Poland; zaborowicz.kasia@gmail.com (K.Z.); ortobb@onet.pl (B.B.); 2Institute of Chemical Technology and Engineering, Faculty of Chemical Technology, Poznan University of Technology, Berdychowo 4, 60-965 Poznan, Poland; zuzanna.buchwald@put.poznan.pl

**Keywords:** digital pantomography, dental age, tooth eruption, permanent canines, premolars

## Abstract

Eruption is a complex and dynamic process determined by both genetic and epigenetic factors. This process involves a number of changes in the tissues surrounding the tooth and in tooth morphology. The aim of this study was to analyze the eruption sequence of permanent canines and premolars on the basis of pantomographic images. The study material consisted of 300 digital pantomographic images of children in the developmental period. The study group consisted of 165 boys and 135 girls. Images of patients of Polish nationality, aged 6–10 years, without diagnosed systemic diseases and local disorders were used in the study. Results: The study has shown that the most common pattern of tooth eruption in the lateral zones is type A positioning of the lateral teeth, which is 4-5-3. This pattern is characteristic of both girls and boys. This pattern also occurs most frequently in the maxilla in both boys and girls. In the mandible, on the contrary, two patterns of lateral tooth eruption were predominant. In girls, types A and E/4-5-3 and 3-4-5/occurred in the mandible, while in boys, types A and C/4-5-3 and 5-4-3/were observed in the mandible. The process of tooth eruption is a recognized measure of a child’s physical development, and pantomographic images are an effective and common diagnostic tool.

## 1. Introduction

Tooth eruption is a physiological process defined as the movement of a tooth from its developmental position in the bone to its functional position in the oral cavity. It is a complex and dynamic process that involves changes in the tissues surrounding the tooth and in the tooth itself. The process of tooth eruption consists of three phases: pre-eruptive, pre-functional and functional [1]. In the pre-eruptive phase, the formation of the crown and its mineralization take place and the formation of the root is initiated. Bone resorption takes place in the area of the eruption path of the tooth. In the pre-functional phase, the tooth changes its position from intra- to extra-periosteal, and after appearing in the oral cavity, it moves toward the contact with the opposing tooth. In this phase, the root is also formed, and changes occur in the bone structure of the alveolar process and other periodontal tissues. The functional phase is the period in which the root continues to grow, the periapical aperture forms and the periapical tissues develop. In this phase, the epithelial attachment of the gingival groove is formed and moves toward the root apex. The depth of the gingival groove decreases, the length of the clinical crown increases, and the process of active eruption of the tooth is completed.

The process of tooth eruption is not continuous but proceeds in stages. There are periods of time between the active movements of the tooth. The average rate of tooth eruption is 0.7 mm per month, which means that the time from the moment the tooth is palpable by the gingival tissue until the complete eruption of the crown is 2 months (fluctuation of 0.9–4.9 months). During the intraosseous stage, the tooth moves at a rate of 1–10 μm per day [2]. Once the erupting tooth reaches the alveolar bone crest and penetrates the soft tissues, the speed of movement changes and reaches 75 μm per day. This stage of tooth development is dependent on local, genetic and epigenetic factors. Scientific reports by Hughes et al. confirm that tooth eruption is genetically programmed and genes are responsible for 70% of it, while environmental factors play a minor role. Studies on rodents have shown that the dental development, number, size, shape and eruption of teeth are under strong genetic control [3].

The mechanism of tooth eruption has not been thoroughly explained. As a result of the analysis of physiological processes, it has been established that the tooth follicle, the periodontium and the membrane covering the periapical tissues, and the term root follicle play an important role in the process of tooth eruption. These structures interact with each other and control the process of tooth eruption [4]. The innervation of the periapical tissue causes a bottom–up pressure on the tooth root surface which pushes the tooth in the eruptive direction, while the follicle destroys the bone tissue located above it, creating an eruptive path. The reduced enamel epithelium is a source of the enzymes collagenases and hydrolases, which cause the fibrous connective tissue to thin and the intercellular matrix to loosen. A properly functioning dental follicle, surrounding the crown of the tooth, is a source of chemotactic cytokines for monocytes, which are precursors of osteoclasts, which are essential for bone resorption. The local factors are colony stimulating factor (CSF-1) monocyte chemotactic protein (MCP-1), epidermal growth factor (EGF), transforming growth factor-beta (TGF-B), parathormone-dependent protein (PTHrP), and bone resorption-stimulating protein interleukin-1 (IL-1). As a result of monocyte transformation and osteoclast activation, a resorption process takes place, resulting in an eruption pathway for the attachment. Parathormone, on the other hand, stimulates bone growth from the base of the bone crypt, resulting in remodeling of the bone and the collagen fibers of the periodontium. These processes are mediated by degrading metalloproteinases (MMP 2, MMP 8) and tissue inhibitors of metalloproteinases [5].

The eruption of deciduous teeth can start as early as 2 or 3 months of age, with the average time for the appearance of the first deciduous tooth being 6.9 ± 0.3 months. The time when all deciduous teeth erupt is between 17 and 25 months. The first teeth to appear in the mouth are usually the lower central incisors, although there is a certain percentage of children whose first teeth in the mouth are the upper central incisors. The timing of the first teething depends on the birth weight, the natural feeding period and the month of birth of the child [6]. In the 2–4 years before the teeth are replaced, the process of resorption of the roots of deciduous teeth begins, which is stimulated by a number of enzymes and proteins that control this process. The main protein involved in the resorption process is receptor activator for nuclear factor and B ligand (RANKL); other factors include colony stimulating factor (CSF-1), monocyte chemotactic protein (MCP-1), transforming growth factor (TGF), osteoprotegerin (OPG), and DNA binding factor (Cbfa) [7]. Resorption starts earlier in mandibular teeth and teeth resorb earlier in girls. Although the root resorption of deciduous teeth is genetically controlled, it also depends on environmental and local factors that may accelerate resorption, such as occlusal trauma, pulp or periodontal inflammation or endodontic treatment. Delayed resorption may be caused by general factors: hypothyroidism, hypopituitarism, or associated with genetic disorders (in genetic syndromes such as hyper Ig-E, di George syndrome, etc.) [8,9].

The eruption of teeth is a physiological process, however, it may deviate from the generally accepted norm. The premature eruption of teeth manifests itself in the presence of innate teeth, when they are already present in the oral cavity at birth. The newborn teeth erupt in the first month of the child’s life. Early teething is the appearance of milk teeth before 5 months of age and permanent teeth before 5 years of age. Disturbances in the eruption of teeth may be a sign of abnormal physical development or may indicate the presence of general or local pathologies in the oral cavity. Abnormal teething may be related to general pathologies such as pituitary or thyroid disorders, infections or nutritional disorders, or it may be caused by local factors such as premature loss of a deciduous tooth, cysts, tooth overabundance or trauma [10,11].

There is an ongoing debate as to whether a change in the order of eruption of teeth is a pathology. Deviations from the accepted pattern combined with abnormalities in the timing of the eruption of teeth or their incorrect position in the arch indicate the existence of abnormalities, which require careful individual assessment. While the sequence of eruption of milk teeth is not significant for the development of the child if the correct number and position in the arch are preserved, the sequence of eruption of permanent teeth is important for the correct development of occlusion. The period of replacement of milk teeth with permanent teeth is a long process and lasts about 6 years. It is a key period in the formation of correct interarch relationships. Careful observation of the process of dental development is an important part of the dentist’s practice. Regular examination supported by X-ray imaging gives the possibility of early dental intervention to eliminate the development of occlusal abnormalities.

Predicting the moment of eruption of permanent teeth is the subject of deliberations of many scientists. Analyses of the development of the alveolar process, the roots of the teeth and the pulp itself are used to determine the time of eruption [12]. Pantomographic imaging is helpful in accurately determining the presence of the pulp and its stage of development [13,14]. In addition, it can be useful for estimating the expected time of eruption of the tooth on the basis of the developmental assessment of the pulp or the analysis of the development of the tooth root [15,16]. This paper describes an analysis of pantomographic radiographs performed to evaluate the eruption sequence of teeth in the lateral dental arches, specifically the permanent canines and premolars. The assessment of the sequence of tooth eruption is an important part of a dentist’s work, allowing the assessment of a child’s development, and a pantomographic radiograph is an effective diagnostic tool in evaluating the pattern of tooth eruption.

## 2. The Aim

The aim of this study was to analyze the eruption sequence of permanent canines and premolars based on the analysis of pantomographic images.

## 3. Materials and Methods

The study material included digital pantomographic radiographs of children in the developmental period. A total of 300 images were analyzed. In the study group, there were 165 boys and 135 girls. Photographs of patients of Polish nationality, aged 6–10 years, without any systemic diseases or lesions and defects in the craniofacial region, without developmental defects of dentition, without diseases of hard tissues of teeth and pulp or trauma were used in the study. The acquired study material was reviewed, and images with developmental abnormalities were excluded. The median age for boys was 8.3 ± 1.2 and for girls was 8.1 ± 1.1. The images were collected from the database of the University Centre for Dentistry and Specialized Medicine in Poznań. The Bioethics Committee of the Poznań University of Medical Sciences, in its decision of 6 October 2021, concluded that the study does not bear the features of a medical experiment and therefore agreed to conduct the research work.

The pantomographic images used in the study were taken with a German Durr Dental camera—VistaPano S Ceph, which is equipped as a standard with an X-ray head with a focus of 0.5 mm, a digital sensor, a Cls-CMOS matrix and additionally enables the images to be taken with S-PAN technology, thanks to which the images have better sharpness and saturation of mineralized structures. The resolution of the X-ray images is 2484 × 1159. Specialized software DBSWIN (Durr Dental) is designed to read digital images recorded by the X-ray machine in DICOM 3.0 format. The application runs in MS Windows. The software allows creating, importing and exporting data and image databases. It reads 16-bit images in 65536 shades of gray and is dedicated for presenting medical radiographs [17].

To analyze the position of the tooth buds in the lateral zones of the maxilla and mandible on the pantomographic images, customized tooth and bone parameters were determined using reference points and lines (the FDI Universal System was used to mark the teeth):

Line X—left vertical edge of the pantomographic image/LX/;

Line L Mx—perpendicular line to the X axis passing through the top of the alveolar process of the maxilla mesially from the tooth examined/LMx/;

Line LMd—the line perpendicular to the X axis passing through the apex of the mandibular alveolar process mesially from the examined tooth/LMd/;

C13—top of canine cusp 13;

C23—top of canine cusp 23;

C33—top of canine cusp 33;

C43—top of canine cusp 43;

C14—top of buccal cusp of tooth 14;

C24—top of buccal cusp of tooth 24;

C34—top of buccal cusp of tooth 34;

C44—top of buccal cusp of tooth 44;

C15—top of buccal cusp of tooth 15;

C25—top of buccal cusp of tooth 25;

C35—top of buccal cusp of tooth 35;

C45—top of buccal cusp of tooth 45.

Measurements were conducted of the distance of the points on the cusps of the permanent canines and cusps of the premolar teeth nearest to them from their perpendicular projections on the LMx line (alveolar process of the maxilla) or on the LMd line (alveolar part of the mandible). Measurements are depicted in Figure 1. Similarly, the distance of these teeth was assessed when the tooth had already erupted. Measurements of the tooth–bone parameters were made with the Open Source ImageJ 1.52a software (LOCI University of Wisconsin) [18]. Patient data with age specified in months were entered into an MS Excel 2013 (Microsoft Excel 15.0.4420.1017) spreadsheet to allow analysis of pantomographic images to be summarized and organized [19]. Next, the tooth and bone parameters measured in ImageJ, determined on panoramic radiographs of children 6–10 years (72–120 months), were entered into the created database. On this basis, statistical data on the position of permanent tooth buds in relation to the alveolar process of the maxilla and the alveolar part of the mandible and erupted teeth were compiled. To analyze the reliability of the measurements, we used the intraclass correlation coefficient in a two-factor mixed model for absolute agreement of single measurement—ICC (intraclass correlation coefficient, two-way mixed model for absolute agreement of single measurement). For all parameters, the agreement of both measurements is statistically significant at the level of *p* < 0.001, and the value of the correlation coefficient r-Pearson was 0.9221–0.9977. Therefore, the reliability of the test by the absolute stability method is high.

## 4. Results

The results were categorized according to the position of the permanent canines, first premolars and second premolars in each quadrant. Six possible patterns of tooth attachment positions were identified (Table 1).

Table 2 presents the results of the study group. Table 3 contains the results of the frequency of occurrence of particular pattern types in boys and girls, and their percentage distribution is shown in Table 4. The study revealed that the most common pattern of tooth eruption in the lateral zones is type A of lateral tooth arrangement, i.e., pattern 4-5-3. This pattern is characteristic of both boys (Figure 1) and girls (Figure 2). This pattern is also most common in the maxilla in both boys (Figure 3) and girls (Figure 4). In the mandible, however, two patterns of lateral tooth eruption predominated. In boys, types A and C/4-5-3/ and /5-4-3/ (Figure 5) were observed in the mandible. In girls, types A and E/4-5-3 and 3-4-5/were observed in the mandible (Figure 6). The percentage distribution of eruption patterns is shown in Figure 7 for boys and in Figure 8 for girls. No significant differences were found in the eruption patterns of the lateral teeth on the right and left sides of the maxilla and mandible in either boys or girls.

Statistical analysis:

For all parameters, the agreement of both measurements (intraclass correlation coefficient, two-way mixed model for absolute agreement of single measurement) is statistically significant at the level of *p* < 0.001, and the value of the correlation coefficient r-Pearson was 0.9221–0.9977. Therefore, the reliability of the test by the absolute stability method is high.

Statistically significant differences between the number of patterns between boys and girls as well as between the quadrants were determined using the χ2 tests. Confidence intervals (CI) for differences in the number of patterns (in percentages) were determined using the interval estimation for the fractions. The alpha significance level was set at 0.05. All calculations were carried out with the use of Statistica 13.1 Software (TIBCO Software Inc. (Palo Alto, CA, USA)).

Statistical analysis indicated that the differences in the frequency of occurrence of patterns A and E in the mandible on the right and left between boys and girls were statistically significant. Statistical analysis confirmed the existence of differences between boys and girls in the frequency of pattern C in the mandible on the right and pattern E in the mandible on the left (Table 5).

Statistical analysis of individual quadrants for boys and girls showed no statistically significant differences (Table 6 and Table 7).

## 5. Discussion

The assessment of the position of permanent tooth buds on a panoramic X-ray is useful in planning orthodontic treatment. It supports the therapeutic management of dental abnormalities in the child during the developmental period. Thanks to this analysis, it is possible to predict the eruption sequence of permanent teeth in lateral sections. A change in the sequence of eruption of teeth may be an indicator of the presence of pathologies in biological development. Changes in the timing of the appearance of teeth are not as important for normal development as the correct sequence of tooth eruption [20]. The sequence of tooth eruption is significantly related to the general development of the child. A comparison of the sequence of eruption of the teeth on the right and left side of the maxilla and mandible is important, and differences in the timing of appearance of the teeth indicate an abnormality. The unicuspid right and left teeth should appear within 4 months, and the prolongation of this period should signal the existence of a developmental defect [21].

The eruption of permanent teeth is divided into three phases. The first phase of dentition is the eruption of incisors and first molars, the second phase is the eruption of premolars and canines, and the third phase is the appearance of the third molar [22]. In the first phase, two possible types of eruption are observed. In the molar type, the first molars appear first in the mouth. In the incisor type, the central lower permanent incisors appear first. Next come the maxillary central incisors, the mandibular lateral incisors (age 7) and the maxillary lateral incisors (age 8) [23,24]. The second phase of the emergence of permanent teeth is characterized by great variability. The authors report a dependence of the eruption sequence on location (maxilla and mandible) and on gender [25].

The replacement of teeth in the lateral sections in the support zones can follow different patterns. In the maxilla, two eruption patterns are described: 4-3-5 and 4-5-3 [26,27].

In the mandible, the order 3-4-5 or 4-3-5 is adopted, which means that the second premolar tooth is erupted last. In our study conducted in the population of Polish children aged 6–10 years, we confirmed the eruption pattern of lateral teeth in the maxilla as 4-5-3. Different eruption patterns of lateral teeth were observed in the mandible. The most common pattern was 4-5-3 in both boys and girls. Only the 3-4-5-5 pattern appeared second in girls. In boys, pattern 5-4-3 appeared as the second. The study was conducted in generally healthy children, with no systemic diseases or lesions or defects found in the craniofacial region, no developmental defects in the dentition, no diseases of the hard tissues of the teeth and pulp or trauma.

In scientific reports, Jarka and Angerman state that the sequence of first premolar, canine and second premolar (4-3-5) is considered to be the correct pattern for the replacement of deciduous teeth with permanent teeth in the maxilla. They also noted that this sequence is more common in girls, while the sequence first premolar, second premolar, and canine is more common in boys (4-5-3) [28].

The sequence of eruption of teeth in terms of right and left predominance is also a predictor of lateralization of the body. The age at which the replacement of teeth begins is the moment when lateralization is finally established, i.e., functional asymmetry, which is related to the dominance of one of the cerebral hemispheres [29]. Abnormal lateralization, or so-called undetermined or crossed lateralization, is related to the immaturity of the nervous system and is a risk factor for dyslexia or the possibility of other learning-related disorders. Attention has been directed to the correlation between tooth replacement and the child’s motor laterality. If the first tooth appears on the right side, the child tends to be right-handed and vice versa. Asymmetrical, non-sequential tooth eruption can be used as a parameter in determining a child’s laterality. The predominant side of the tooth eruption order is related to the occurrence of predominance of one side of the body, so determining it can be helpful when working with children with crossed or unclear laterality. The assessment of tooth eruption order can be a useful predictive test for pediatricians and educators when working with children. In our study, we did not observe any significant abnormalities in the order of eruption of the lateral teeth on the right and left side.

Non-consecutive tooth replacement in the lateral segments may be a symptom of a disorder involving the second premolar [30]. This tooth shows the greatest developmental deviations. The most frequently observed is its delayed formation or change in the direction of eruption.

One of the causes of disorders in the sequence of tooth eruption is also the premature loss of a deciduous tooth, which may occur due to trauma or caries and its complications. After the loss of a deciduous tooth, the growth of the alveolar bone in the area of the missing tooth may be inhibited, and consequently, the teeth in the vicinity of the gap may be displaced, tilted or rotated. All these changes lead to occlusal abnormalities. Therefore, it is necessary to take preventive measures [31]. If it is necessary to remove a deciduous tooth because of inflammation, it is advisable to conduct a proper radiological examination and evaluate the presence of the permanent teeth and the position of the teeth buds and the prognosis of their appearance in the oral cavity.

The technological development observed in dentistry creates the possibility of creating a tool that would perform a quick and precise assessment of the position of tooth buds in the bone and would indicate the timing and sequence of tooth eruption. Such a tool is artificial intelligence, whose application in orthodontics has proven to be a reliable and time-saving technique. The data obtained from medical examinations are the basis for the construction of a neural model, which can provide accurate results regarding diagnosis, help in the interpretation of images, and forecast treatment results. A well-trained artificial intelligence model can help identify landmarks, linear and angular measurements and, through automation, improve the doctor’s work [32].

## 6. Conclusions

The process of tooth eruption is a recognized measure of a child’s physical development. Disorders may indicate abnormal development or the presence of general or local pathologies in the oral cavity. In some cases, teething disorders are the first clinically detectable symptom of systemic pathology. Therefore, it is extremely important to know the physiology of this process, the factors that may disrupt its course and the careful observation of the teething process by pediatricians and pediatric dentists [2]. Pantomographic radiograms are an effective and common diagnostic tool in the assessment of the pattern of tooth eruption.

## Data Availability

The study was not publicly funded. Data are not open-ended.

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
