# Peer review of "Eruption Pattern of Permanent Canines and Premolars in Polish Children"

_ijerph, 2022, doi:10.3390/ijerph19148464_

Round 1
Reviewer 1 Report
Dear authors,
The paper provides important epidemiological data regarding eruption pattern in polish children and it will definitely find interest among IJERPH readers. However, it is not suitable for publication in present form. Please introduce following corrections:
The introduction is written in depth and thoroughly. However it should be within the BACKGROUND SECTION together with aim of the study (please look in the instruction for authors on IJERPH website).
Please add the reference number of your local committee decision after the conclusions. Even if your study was exempt from the evaluation You should have received a proper number.
Please provide in brackets the manufacturers of the hardware and software used.
Please provide the inclusion criteria of patients included in the study. Please explain how the study group was designed.
Please provide standard deviations for patients characteristics and try to elaborate wherever there are some differences within the groups.
Were the panoramic x-rays examined only once? If so, please re-examine them and perform error study. Please provide intraclass correlation coefficient (ICC) and Dahlberg’s coefficient. Otherwise, the study won’t be acceptable in any form.
Please change thoroughly the references. You cited mainly the handbooks. The textbooks are mainly a source of professional, not scientific knowledge. The citations 1 and 8 are the same. Please replace the citations of handbooks with the articles on which they were basing. Leave the citations of handbooks only if the other source of information remains unknown.
Author Response
Thank you very much for all the valuable comments on the text. The relevant changes have been made as indicated. The research material consisted of 300 digital pantomographic radiographs of children aged 6-10 years collected from the patient database of the University Center for Dentistry and Specialist Medicine in Poznań. Children with developmental disorders and abnormalities were excluded from the study. All patients were Polish citizens. Due to the nature of the study, the age group consisted of patients aged 6-10 years, in whom it was possible to assess the position of the buds on the radiograph and the order of eruption of teeth.
The study was not a medical experiment, and the Bioethics Committee of the Medical University of Poznan agreed to use radiographs for analysis. I enclose a pdf file with the decision of the committee.
The reference list has been changed and references to textbooks have been replaced with articles.
Regards Katarzyna Cieślińska.
Best Regards Katarzyna Cieślińska

Reviewer 2 Report
great work!!!
Author Response
Thank you very much for your positive review.
Reviewer 3 Report
This study design was a cross-section study based on 300 X-ray images. Therefore, the determination of the eruption sequence seemed to be challenging. The demography of the age and gender should be described. The definition of tooth eruption and the sequence was not very clearly stated in the method. Including a typical X-ray image for each eruption pattern would be helpful.
The software version of data processing should be included. The resolution or pixel size of the X-ray images should be described. Please described the tooth number system used in this manuscript.
In table 1, adding captions “Pattern Code” and “Sequence of Tooth Eruption” would be good. I would suggest merge tables 3 and 4 together to help the reader know the percentile immediately. Also adding columns showing the total percentage for all the quadrilles would be better.
Please add statistical analysis on the difference between boys and girls and among the quadrilles.
Author Response
Thank you very much for all your valuable comments on the manuscript. Appropriate corrections have been made as suggested. I allowed myself to leave tables 3 and 4 in their original form because combining them made it less readable.
Reviewer 4 Report
Overall, I think the study was very well done and presented at a high level. I gained a great deal of information from it. Excellent job. I have some comments, suggested changes and questions. While many are related to English, grammar, sentence structure, there are a few related to references and other topics.
Eruption pattern of permanent canines and premolars in polish children Review
Title:
Eruption pattern of permanent canines and premolars in polish children SHOULD READ Eruption pattern of permanent canines and premolars in Polish children
Abstract:
Line 15 - tooth eruption in the lateral zones is type A positioning of the lateral teeth, which : Question - what are the "lateral zones”? Are those the teeth in the canine and premolar region only? What about the molars? Are they not in the same “lateral zone”?
Line 52 - The mechanism of tooth eruption has not been thoroughly explained so far: Does this mean has not been explained as of this point in the paper or as of June 2022? I think that so far should be removed.
Line 67 - tion-stimulating protein IL-1. SHOULD READ tion-stimulating protein IL-1 (Interleukin 1).
Lines 137 – 139 - additionally enables the take of images in S-PAN technology, thanks to which the images have better sharpness and saturation of mineralised structures. SHOULD READ enables the images to be taken with S-PAN technology. This enables the images to have better sharpness and saturation of mineralized structures. QUESTION – Is there a reference to back this up? That is a pretty important statement to not have one.
Lines 171 – 173 - Patient data with age specified in months were 171 entered into an MS Excel spreadsheet. This is a tool from the MS Office family of programs 172 that allows the data obtained from the analysis of pantomographic images to be summa-173 rised and organized [18]. SHOULD READ Patient data with age specified in months were entered into an MS Excel spreadsheet to allow analysis of pantomographic images to be summarized and organized [18].
Line 239 - in planning the orthodontic treatment SHOULD READ in planning orthodontic treatment.
Lines 239 – 240 - and supports the therapeutic management of 239 dental abnormalities in the child during the developmental period. This should be a separate sentence starting with It supports. Question – are you only discussing abnormalities of the teeth (strictly dental) and not dento-facial? If that’s the case, I am OK with it as I have suggested in 2 sentences.
Lines 255 – 256 - In the incisor type, the medial lower permanent incisors appear 255 first. Next come the maxillary central incisors, the mandibular lateral incisors . Here is my question. There are times you refer to central incisors as medial incisors, other times as central incisors. They should be referred to as central incisors throughout the paper. There is an earlier reference to this which will need to be changed as well.
From lines 276 – 284 there are statements made about the side in which teeth erupt first related to “handedness”, delayed eruption having CNS or intellectual related consequences. These statements ALL need references unless they are all covered by reference # 29. If they are covered by that reference, it should be listed first with a statement to the effect that the related issues and findings have been reported in this document.
Line 298 - it is advisable to take a panoramic photo and evaluate. First of all, a panoramic is a radiograph. Second, it is significantly less radiation to take a periapical radiograph, especially if you are only looking for the presence, absence, location and developmental state of one permanent tooth.
Author Response
Thank you very much for all your valuable comments on the manuscript. Appropriate revisions have been made as indicated. Detailed information about the S-PAN technology has been included in the references.
For this study, the eruption pattern of permanent canines and premolars was analyzed; molars were not considered because this was not the purpose of the study.
The analysis of bracket alignment applies only to dental abnormalities; for dental-facial abnormalities, other points on the radiograph would need to be worked out.
Best regards Katarzyna Cieślińska
Round 2
Reviewer 1 Report
Dear authors,
You have signficatly improved Your manuscript. However, there are till some issues to be corrected.
In the line 65-69, first full name, then the shortcut in the brakcets.
[For all parameters the agreement of both 197 measurements is statistically significant at the level of p<0.001 and the value of the corre- 198 lation coefficient r-Pearson was 0.9221-0.9977. Therefore the reliability of the test by the 199 absolute stability method is high.] - this part should be moved to results section to the special paragraph.
Citations 16-19 should be removed. This is not a source of knowledge. It is enough to mention the manufacturer in brackets within the text.
You still did not mention the limitations of the study in discussion.
After "Conclusions" You should provide information as:
Supplementary Materials - please add the document from bioethical committe You sent me as cover letter and name it as Supp. 1. You can add Supplementary material while resubmitting the manuscript.
Funding
Author Contributions
Institutional Review Board Statement: Here write: The study was exempt from ethical approval by the ethical committe of University of Poznan... (Suppl 1)
Conflicts of Interest
Data Availability Statement
Informed Consent Statement
Everything in accrodance with instructions for authors.
PLEASE READ CAREFULLY THE INSTRUCTIONS FOR AUTHORS on IJERPH website.
Only after introducing all of the above mentionted corrections, the study can be accepted for publication.
Remeber to adress every issue to the reviewer in form of the answers to every paragraph of the review SEPARETLY.
Author Response
Dear Reviewer
Thank you very much for your comments. All corrections have been made as indicated.
In lines 65-69, abbreviations have been placed in brackets.
A paragraph on statistical analysis was inserted and the relevant paragraph was moved.[ For all parameters the agreement of both measurements is statistically significant at the level of p<0.001 and the value of the correlation coefficient r-Pearson was 0.9221-0.9977. Therefore the reliability of the test by the absolute stability method is high.]
I have taken the liberty of leaving the quotation 16-19 as I think it may be useful for other authors.
Limitations information was introduced in the discussion as suggested.[ The study was conducted in generally healthy children, with no systemic diseases or lesions or defects found in the craniofacial region, no developmental defects in the dentition, no diseases of the hard tissues of the teeth and pulp or trauma.]
I have inserted all the necessary information after the conclusion.
I will attach the bioethics committee's consent document as a supplement.( I have sent it to the editor already)
Once again, my sincere thanks for the correction.
Katarzyna Cieślińska